# TLR Agonists as Vaccine Adjuvants Targeting Cancer and Infectious Diseases

**DOI:** 10.3390/pharmaceutics13020142

**Published:** 2021-01-22

**Authors:** Marina Luchner, Sören Reinke, Anita Milicic

**Affiliations:** 1Department of Biochemistry, Magdalen College Oxford, University of Oxford, Oxford OX1 4AU, UK; marina.luchner@magd.ox.ac.uk; 2The Jenner Institute, Nuffield Department of Medicine, University of Oxford, Oxford OX3 7DQ, UK; soren.reinke@ndm.ox.ac.uk

**Keywords:** Toll-like receptors, adjuvant, vaccine, immunotherapy, cancer, immunogenicity, MyD88, NF-κB, type I interferon

## Abstract

Modern vaccines have largely shifted from using whole, killed or attenuated pathogens to being based on subunit components. Since this diminishes immunogenicity, vaccine adjuvants that enhance the immune response to purified antigens are critically needed. Further advantages of adjuvants include dose sparing, increased vaccine efficacy in immunocompromised individuals and the potential to protect against highly variable pathogens by broadening the immune response. Due to their ability to link the innate with the adaptive immune response, Toll-like receptor (TLR) agonists are highly promising as adjuvants in vaccines against life-threatening and complex diseases such as cancer, AIDS and malaria. TLRs are transmembrane receptors, which are predominantly expressed by innate immune cells. They can be classified into cell surface (TLR1, TLR2, TLR4, TLR5, TLR6) and intracellular TLRs (TLR3, TLR7, TLR8, TLR9), expressed on endosomal membranes. Besides a transmembrane domain, each TLR possesses a leucine-rich repeat (LRR) segment that mediates PAMP/DAMP recognition and a TIR domain that delivers the downstream signal transduction and initiates an inflammatory response. Thus, TLRs are excellent targets for adjuvants to provide a “danger” signal to induce an effective immune response that leads to long-lasting protection. The present review will elaborate on applications of TLR ligands as vaccine adjuvants and immunotherapeutic agents, with a focus on clinically relevant adjuvants.

## 1. Historical Perspective

The discovery of Toll-like receptors (TLRs) is considered as one of the major landmark findings in immunology research, honoured with the 2011 Nobel Prize in Physiology or Medicine alongside the discovery of dendritic cells (DCs) [1]. While today there are numerous reports on applications of TLR agonists, their mechanisms of action remain the subject of intensive research. The era of discovery was heralded by Charles Janeway’s address at the 1989 Cold Spring Harbor Symposium, when he predicted that pattern-recognition receptors (PRRs) are the link between innate and adaptive immunity [2]. The first mammalian receptor to fulfil Janeway’s requirements for a PRR was found in 1994 and termed hToll because of its resemblance to Toll, a *Drosophila melanogaster* cytosolic protein domain. Innate hToll signalling in antigen-presenting cells (APCs) was observed to link innate and adaptive immunity through the expression of the gene encoding CD80, which co-stimulates T cells. In the following years, additional Toll homologues were discovered and named Toll-like receptors (TLRs), which led to the renaming of hToll into TLR4 [1].

In 1998, it was shown that TLR4 recognises bacterial lipopolysaccharide (LPS), and transmits the inflammatory signal across the cell membrane [3]. This was confirmed by Akira and colleagues, who demonstrated that TLR4-knockout mice failed to respond to LPS [4], rendering it the first TLR agonist to be discovered in vivo. Next, cytidine-phospho-guanosine (CpG) DNA was identified at the turn of the century as an agonist to TLR9, by demonstrating that TLR9-deficient mice do not show any immune response to CpG DNA [5]. In 2002, similar knock-out experiments demonstrated that imidazoquinolines act on TLR7 through a MyD88-dependent signalling pathway [6]. It subsequently took ten years to reveal that TLR8 is able to sense bacterial RNA (in phagosomal vacuoles), and single-stranded viral RNA. The reason for the delayed TLR8 discovery lies in the fact that TRL8 was originally thought to be non-functional. On the contrary, the TLR8 mechanism is even more complex than the signalling pathways of its relatives since it includes cross-talk with other endosomal TLRs [7]. TLR8 is furthermore implicated in the generation of autoimmunity.

Recent studies investigated the connection between TLRs and autoimmune or neurodegenerative diseases. While TLR2, TLR4, TLR5 and TLR10 might be involved in the pathogenesis of systemic sclerosis [8], TLR activation can have both destructive and protective effects on Alzheimer´s disease (AD) [9]. Furthermore, multiple genetic TLR polymorphisms have been identified as protective or risk factors for AD [9]. Additionally, there is evidence that TLR2 and TLR4 play a role in Parkinson´s disease through microglia activation [10,11,12,13,14]. Thus, research on TLRs in health and disease is still an emerging field with high potential for clinical applications.

The underlying signalling pathways downstream of mammalian TLRs have largely been resolved over the past 15 years (Figure 1). Upon binding of the appropriate ligand, TLRs recruit TIR domain-containing adaptor proteins like MyD88, which activates NF-κB signalling, and mitogen-activated protein (MAP) kinases, which in turn initiates the expression of pro-inflammatory cytokines such as tumour necrosis factor α (TNF-α) or Interleukin 6 (IL-6) [15]. Alternatively, or in addition, TLRs can activate interferon regulatory factor (IRF)-mediated type I interferon (IFN) responses, e.g., IRF3 and IRF7, which play an essential role in bridging innate and adaptive immunity through the induction of co-stimulatory molecules on APCs [16,17].

In order to turn knowledge into application, researchers have aimed to take advantage of the fact that TLRs in general activate the innate immune system, which subsequently leads to effective adaptive immunity. With the rise of subunit vaccines, which show significantly lower immunogenicity than whole attenuated pathogens, the necessity to add adjuvants became pressing. The first TLR agonist to become approved for use as a vaccine adjuvant is Monophosphoryl lipid A (MPL), a detoxified LPS derivative that activates TLR4 (Figure 1). MPL, together with an aluminium salt is part of the adjuvant system AS04, which was licensed as a component of the Cervarix vaccine in 2009 [18]. Approval of the TLR9 agonist CpG oligodeoxynucleotides (ODN) 1018 as an adjuvant for the vaccine Heplisav-B followed in 2017 [19]. These two licensures led to a wave of investigations of a variety of TLR agonists, which will be further described below.

## 2. TLR3 Agonists

### Poly-IC and Derivates

Acting on both TLR3 and RIG-I-like receptors (RLRs), the synthetic dsRNA molecule polyinosinic-polycytidylic acid (poly-IC) not only triggers the innate immune response, but also induces subsequent activation of adaptive immunity. As a reaction to disappointing results of an early Phase I/II clinical trial using poly-IC in patients with leukaemia [20], other variants of poly-IC were developed in order to improve safety, interferon induction, and immunogenicity of the parent molecule. Such poly-IC derivates include poly-IC12U (Ampligen^®^, Hemispherx) and poly-ICLC (Hiltonol^®^, Oncovir). Poly-IC12U represents a modified variant of poly-IC containing mismatched uracil and guanosine residues, whereas poly-ICLC contains poly-L-lysine in carboxymethylcellulose, which increases resistance against nucleases and thereby enhances and prolongs stability [21].

Poly-IC12U has been evaluated as a treatment against infectious diseases: in the 1990s, it was first tested as a therapeutic agent against human immunodeficiency virus (HIV) type 1 infection and showed promising results [22]. In patients with chronic fatigue syndrome/myalgic encephalomyelitis (CFS/ME), treatment with the poly-IC12U drug rintatolimod led to an improvement in exercise tolerance and showed dose-sparing effects in a Phase III clinical trial in 2012 [23]. However, the efficacy of the agent seems to be controversial as a review on the treatment of CFS/ME suggested that trials of rintatolimod provide only low to moderate strength of evidence [24].

In addition, poly-IC12U was tested as an adjuvant for cancer vaccines: intraperitoneal poly-IC12U (Ampligen^®^) in combination with a DC-based vaccine (αDC1) is currently being evaluated for treatment against ovarian cancer and peritoneal malignancies [25]. Poly-IC12U exclusively activates TLR3, thereby potently inducing maturation of DCs, with a sustained production of IL-12 which drives Th1 responses [26].

Clinical trials also investigated the related molecule Poly-ICLC as an adjuvant for cancer vaccines. Intramuscular administration of Poly-ICLC appears to reduce toxicity and was implicated in prolonged survival with tumour stabilisation or regression of patients with anaplastic astrocytomas and glioblastomas [27]. In addition, subcutaneous administration markedly increased the immunogenicity of a peptide vaccine in patients with ovarian cancer in a Phase I clinical trial [28]. Thereby, the inclusion of poly-ICL accelerated seroconversion, increased antibody titres, broadened the repertoire of antibodies and supported CD8^+^ and CD4^+^ T cell induction [28]. TLR3 activation by Poly-ICLC signals through a clear Th1 phenotype, characterised by the induction of IL-12, TNF-α, IFN-γ, IL-6 and type I interferon, as well as the chemokines KC, MCP1, MIP1-α and MIP1-β [21].

## 3. TLR4 Agonists

### 3.1. Monophosphoryl Lipid (MPL)

As a result of its early discovery as a TLR4-stimulating adjuvant, MPL became the first TLR agonist to gain approval for use in humans as part of vaccines against infectious diseases. In the form of the AS01 adjuvant system, combined with liposomes and saponin, it is included in the herpes zoster vaccine Shingrix [29]. Adsorbed to aluminium salts within the AS04 adjuvant, it is part of the human papillomavirus vaccine Cervarix [30,31] and the hepatitis B vaccine Fendrix [32,33]. Clinical trials on Shingrix especially illustrate the benefits of MPL as an adjuvant, demonstrating robust protection against herpes zoster in older adults and efficacy in immunocompromised individuals [34].

MPL is not exclusively used against viral pathogens. A subcutaneously administered four-injection immunotherapy against allergic rhinitis (Pollinex Quattro) consisting of specific allergoids and MPL has shown clinical efficacy and good tolerability in children. The benefits of MPL in anti-allergic treatments are based on increased allergen-specific IgG titres and dampened IgE response during allergen exposure [35].

Activation of TLR4 through MPL triggers NF-κB, leading to the production of proinflammatory cytokines, such as TNF-α and IL-6. These cytokines enhance the adaptive immune response through stimulating APC maturation in parallel with the ability to inhibit regulatory T cells and supress tolerance [36]. MPL is generally reported to enable skewing towards the Th1 immune response through enhanced IFN-γ production by antigen-specific CD4^+^ T cells [36].

### 3.2. Glucopyranosyl Lipid A in a Stable Emulsion (GLA-SE)

In order to expand the beneficial effects of MPL, the synthetic TLR4 agonist Glucopyranosyl Lipid A in a stable emulsion (GLA-SE) was developed.

GLA-SE was first evaluated as an adjuvant against infectious diseases. Created to enhance influenza vaccines, GLA-SE performed well as an adjuvant for the H5N1 influenza vaccine in a Phase II clinical trial [37]. In addition to improving the humoral response, it also demonstrated a dose-sparing effect, which is especially beneficial in a pandemic situation [37]. In other clinical trials with a positive outcome, GLA-SE was tested as an adjuvant with the ID93 tuberculosis (TB) vaccine based on four protein antigens [38].

Moreover, GLA-SE was shown to be an effective agent for the active immunotherapy regimen against recurrent soft tissue sarcoma expressing the NY-ESO-1 tumour antigen (CMB305) [39] and as intra-tumoural treatment in patients with Merkel cell carcinoma [40].

GLA is optimised for binding to the human MD2 molecule of the TLR4 complex and has been shown to act on APCs, resulting in the induction of cytokines such as IFN-γ, TNF and IL-2. Addition of GLA-SE to the influenza vaccine Fluzone increased antibody and T cell responses and broadened the serological specificity in comparison to Fluzone alone [41].

## 4. TLR5 Agonists

### 4.1. Mobilan

The adenovirus Ad5-vector-based vaccine Mobilan was developed for intra-tumoural delivery with the intention to extend the application of TLR5-targeting anticancer immunotherapy to tumours that do not naturally express TLR5. The vector drives the expression of a self-activated TLR5 signalling cassette, which comprises human TLR5 and a secreted derivative of the natural TLR5 ligand flagellin from *Salmonella* species [42]. In 2020, the first completed clinical study of Mobilan as a drug candidate for immunotherapy in prostate cancer patients demonstrated safety, tolerability and elevation of prostate-specific antigens and cytokine levels and increased lymphoid infiltration into prostate tissue [43]. Mobilan was thereby capable of maintaining strong NF-κB signalling and led to the recruitment of innate immune cells, including neutrophils and NK cells. Of note, Mobilan induced enhanced NF-κB activation compared to the TLR5 agonist entolimod.

### 4.2. Entolimod

Entolimod is a recombinant protein, a pharmacologically optimised flagellin derivative. In numerous murine models, it has shown anti-tumour effects, initiated through the CXCR3-dependent NK-DC-CD8^+^ T cell axis [44]. A Phase I clinical trial corroborated these results, demonstrating that entolimod induces plasma cytokines including G-CSF, IL-6, IL-8 and IL-10 with no indications of a cytokine storm or the development of neutralising antibodies [45]. Interestingly, entolimod showed radioprotective activity in murine and primate models without reducing tumour radiosensitivity [44]. Therefore, entolimod could not only be a versatile player in cancer therapy, especially in combination with radiotherapy, but also be used as a radiation countermeasure for acute radiation syndromes. As a result, the FDA has granted an investigational new drug, fast-track and orphan drug status for entolimod [46].

## 5. TLR7 Agonists

### 5.1. Resiquimod

Resiquimod, a TLR7/8-bispecific agonist, is a prototypical imidazoquinoline molecule [47]. In vitro studies have demonstrated that resiquimod stimulates DC maturation by inducing a Th1 cytokine profile and enhancing costimulatory molecule expression. This leads to a more efficient cross-presentation of exogenous antigens and stronger antigen-specific CD8^+^ T cell responses [48].

Resiquimod was first tested in mice as an adjuvant for vaccines against infectious diseases, such as influenza [47]. However, early studies showed that soluble molecules like resiquimod diffuse quickly from the site of injection throughout the body and fail to induce local immune activation, thereby limiting its clinical applicability [47].

In addition, resiquimod was evaluated as an adjuvant for cancer vaccines. A clinical trial in patients with high-risk melanoma tested the NY-ESO-1 protein adjuvanted with resiquimod emulsified into a water-in-oil adjuvant Montanide, which can serve as the depot at the site of injection [48]. The trial revealed good safety and antibody immunogenicity with enhanced CD4^+^ T cell responses. However, this adjuvant combination was not sufficient to induce consistent CD8^+^ T cell responses against NY-ESO-1.

### 5.2. Imiquimod

The synthetic molecule imiquimod superficially resembles a nucleoside analogue but lacks the fourth nitrogen atom present in purines, and instead contains an imidazoquinoline and isobutyl group [49]. Binding of TLR7 by imiquimod induces secretion of pro-inflammatory cytokines, predominantly IFN-α, TNF-α and IL-12. This creates a local cytokine milieu biased towards a Th1-type response, with the generation of cytotoxic effectors [49].

Apart from the EU-authorised clinical use of topical imiquimod (Aldara^®^ cream containing 5% imiquimod, by Meda Pharmaceuticals), which is indicated for external genital warts, actinic keratosis and superficial basal cell carcinoma [50], there are no TLR7-adjuvanted vaccines that have progressed beyond early-phase clinical studies to date. One reason for this stagnation might be the rapid diffusion of imiquimod away from the vaccination site, as mentioned above, and therefore away from the antigen, which diminishes its adjuvant activity. A suggested approach to solve the drug distribution issue has been the topical application of imiquimod to the skin alongside conventional vaccination [47].

Imiquimod was further evaluated as an adjuvant for cancer vaccines. Clinical trials testing topical imiquimod as an adjuvant for immunisation with the NY-ESO-1 antigen in patients with malignant melanoma [51] and as treatment of breast cancer skin metastases [52] have shown good tolerability and a systemic immune response [51,52].

## 6. TLR9 Agonists

### 6.1. CpG

TLR9 activation by CpG recruits MyD88, IL-1R associated kinase (IRAK) and TNF receptor-associated factor 6 (TRAF6). This leads to the activation of several mitogen-activated kinases and transcription factors, such as NF-κB and AP-1, resulting in transcription and secretion of proinflammatory chemokines and cytokines [53]. In order to mimic the immunostimulatory effect of bacterial DNA on TLR9, synthetic ODNs containing unmethylated CpG dinucleotides have been developed [53,54].

At least three distinct classes of CpG ODN have been identified based on differences in structure and mechanism of the immune response: (i) K-type ODNs (also known as B-type), (ii) D-type ODNs (also referred to as A-type), and (iii) C-type ODNs [53]. All sub-classes of ODNs characteristically trigger immune responses through the activation of plasmacytoid DCs (pDCs) and induction of cytokines such as TNF-α and IFN-α [53]. Furthermore, ODNs can enhance antibody responses [55]. Of note, structural differences in the individual ODN-classes (reviewed by Vollmer et al. [56]) feature different characteristics that may be relevant for vaccine development. For example, K-type ODNs, which carry 1 to 5 CpG motifs on a phosphorothioate backbone display improved resistance to nuclease digestion and an extended in vivo half-life [53].

Different CpG variants were tested as vaccine adjuvants against infectious diseases. One success story is the use of CpG 1018 as an adjuvant for hepatitis B virus (HBV) vaccine Heplisav-B, developed by Dynavax Technologies Corporation [19]. Clinical studies of vaccines based on the hepatitis B surface antigen consistently demonstrate more rapid induction of protective antibody titres when adjuvanted with CpG 1018 compared to aluminium salts. This has been observed in different population groups, including the elderly and immunocompromised individuals [19]. Similar encouraging results were documented in a randomised, double-blind controlled trial testing the immunostimulatory effect of CPG 7909 as an adjuvant to the HBV vaccine Engerix-B. The addition of CPG 7909 achieved rapid, higher, and sustained seroprotection and increased HBV-specific T helper cell responses compared to Engerix-B alone [57]. In addition to use with HBV, there are also applications of CpG in malaria vaccines. In a Phase I clinical trial of the protein-based malaria vaccine BSAM2/Alhydrogel, addition of CpG 7909 was shown to have a dose-sparing effect [58]. This feature would especially be of value in resource-poor settings. Another malaria vaccine, BK-SE36, comprising a recombinant protein (SE36) adjuvanted with aluminium hydroxide gel and CpG, has recently demonstrated an acceptable safety profile and enhanced immunogenicity in a small Phase I trial [59].

### 6.2. MGN1703

MGN1703 is a novel, synthetic, covalently closed DNA molecule, which activates the TLR9 signalling pathway.

This molecule was tested as an immunotherapeutic agent against cancer. A randomised, double-blind, clinical trial in patients with metastatic colorectal carcinoma showed that MGN1703 was generally well tolerated, with mostly mild or moderate side effects [60].

Furthermore, MGN1703 was evaluated as an adjuvant for the treatment against infectious diseases. Clinical studies revealed that MGN1703 application increases B cell differentiation in patients receiving MGN1703 concurrent with antiretroviral therapy [61] and that MGN1703 induces a potent type I IFN response in the absence of general inflammation in patients receiving suppressive antiretroviral therapy [62].

Structural and preclinical studies have shown that MGN1703 is a potent TLR9 agonist with limited capacity for interactions with molecules outside its target structure [60]. The effect of MGN1703 on cellular activation was demonstrated by a significantly increased expression of specific surface activation markers, such as CD86, CD40, HLA-DR, CD169 and CD69 and an increased release of a number of cytokines and chemokines such as IFN-γ, IFN-α, IL-6 and IL-8 [63].

### 6.3. SD-101

SD-101 is a synthetic CpG ODN that stimulates pDCs through TLR9 engagement, which causes them to release IFN-α and mature into efficient APCs [64]. Due to its immunostimulatory effect, SD-101 was tested as an agent for anti-cancer immunotherapy. Preclinical studies in mice demonstrated that a combination of intratumorally injected SD-101 combined with systemic PD-1 blockade led to a complete, durable rejection of all injected tumours and the majority of distant-site tumours [65]. Consistent with these findings, in a multicenter Phase I/II clinical trial, direct injection of SD-101 into low-grade non-Hodgkin’s lymphoma in combination with low-dose radiation was well tolerated and resulted in tumour regression [66]. An increase in effector CD8^+^ and CD4^+^ T cells, along with a reduction in regulatory and follicular helper T cells, was observed in the tumour microenvironment.

### 6.4. IC31

In order to combine the immunostimulatory effect through both TLR9 and TLR3 activation, ODNs based on poly-IC were designed, creating a novel single-stranded molecule ODN1a, consisting of dimeric repeats of deoxy-Inosine/deoxy-Cytosine linked by an unmodified phosphodiester backbone [67]. ODN1a was further combined with the antimicrobial peptide KLKL_5_KLK, creating the promising novel adjuvant IC31, which signals via the TLR9/MyD88-dependent pathway of the cellular and humoral immune response. Studies in mice demonstrated that IC31 helped to induce potent antigen-specific cytotoxic T cells and strong protein-specific humoral responses [67]. Furthermore, stimulation of murine bone marrow-derived DCs by IC31 enabled an enhanced T cell proliferation and differentiation [67]. Unlike some CpG ODNs, which can result in systemic side effects, there was no production of pro-inflammatory cytokines, such as IL-6 and TNF-α [67].

Clinical trials investigated the benefit of IC31 as a vaccine adjuvant against infectious diseases. In this context, IC31 has been used as an adjuvant in the candidate tuberculosis vaccine H56:IC3, a novel subunit vaccine consisting of a triple antigen (Ag85B, ESAT-6 and Rv2660c) fusion protein formulated in IC31. The first clinical trial phase I of H56:IC3 demonstrated tolerability and immunogenicity, inducing antigen-specific IgG and persistent H56-specific CD4^+^ T cell responses. In contrast, the efficacy of H4:IC31, another IC31-adjuvanted vaccine against sustained *M. tuberculosis* infection, based on a fusion protein of the antigens Ag85B and TB10.4, did not meet standard statistical criteria in a randomised, controlled Phase II clinical trial [68].

## 7. Vaccines that Work as TLR Agonists

### 7.1. BCG Vaccine

Bacillus Calmette-Guérin (BCG) is an attenuated strain isolated from *Mycobacterium bovis*, first used as a vaccine against TB in 1921 [69]. Although its efficacy varies greatly in different settings it is still widely used as a vaccine against TB and also leprosy.

In the early 1970s, BCG was suggested to have anticancer potential, following a number of reports of tumour growth and metastasis prevention, and tumour regression in mice after BCG administration [70]. Early multicentre case-control studies showed a significant correlation between a reduced risk of melanoma and BCG vaccination in early childhood, and prolonged survival of patients with malignant tumours after surgical management [71,72]. Consistent with this, in a Phase III clinical trial of combinatorial immunotherapy with intralesional BCG and topical 5% imiquimod cream, 56% of the patients had complete regression of in-transit melanoma, where the tumour cells spread via the lymphatic system [73]. However, in another phase III randomised trial of BCG as adjuvant therapy for melanoma, no benefit of BCG was observed [74]. Less controversial are the results regarding BCG immunotherapy against bladder cancer. BCG has been proven to be more effective than chemotherapy in several trials [75]. Nevertheless, the response to BCG remains unpredictable, and severe side-effects occur in around 5% of the patients [76].

In line with mycobacterial infection, TLR2, -4 and -9 might also play a role in the immune response to BCG [77]. TLR recognition of mycobacteria results in activation and nuclear translocation of transcription factors such as NF-κB, leading to the production of nitric oxide and cytokines such as TNF-α, IL-1β and IL-12. In addition, mycobacterial components are recognised by TLR2 and TLR4, leading to neutrophil release of TNF-related apoptosis-inducing ligand (TRAIL) which is required for the maturation of DCs [77].

### 7.2. RNA-Vaccines

Whereas conventional mRNA vaccines consist of an open reading frame encoding the targeted antigen flanked by untranslated regions and a terminal poly(A) tail, self-amplifying mRNA vaccines contain an engineered part of the viral genome encoding the RNA replication machinery in addition to the vaccine antigen [78].

An important advantage of mRNA vaccines is their ability to act as self-adjuvating agents by stimulating innate immunity [78]. In addition to encoding the antigen, the RNA molecule also triggers innate PRRs, such as TLRs and RIG-I-like-receptors. An in vivo study in mice and a human in vitro model evaluated the adjuvant effect of an mRNA-based vaccine encoding influenza A hemagglutinin of the pandemic strain H1N1pdm09 [79]. The findings suggest that mRNA vaccines activate cellular RNA sensors in humans and in mice by upregulating the genes encoding RLRs, TLRs and C-type lectin receptors [79]. The immunostimulatory feature of mRNA vaccines may provide an adjuvant activity to drive DC maturation, thus inducing resilient T and B cell responses [80]. The other side of the coin is that strong induction of type I IFNs and inflammatory cytokines by mRNA vaccines through TLRs and other RNA sensors could have a negative impact on the expression of the vaccine antigen [80,81]. Thus, mRNA vaccine formulations should provide a good balance between antigen expression and adjuvant effect in order to achieve optimal antigen production and innate immune activation [82]. It is important to note that this IFN induction is dependent on the quality of the in vitro transcribed (IVT) mRNA, the delivery vehicle and the administration route [78].

Experimental applications of mRNA vaccines have ranged from cancer therapy to the prevention of infectious diseases. In cancer therapy, mRNA vaccines express tumour-associated antigens that trigger cell-mediated immune response to clear or inhibit cancer cells [78]. The first Phase I/IIa clinical trial in prostate cancer patients revealed tolerability and immunogenicity of a self-adjuvanted RNActive^®^ vaccine CV9103, which activates, among others, TLR7 [83].

Clinical development of mRNA vaccines was suddenly accelerated with the SARS-CoV-2 outbreak [84]. In 2020, BioNTech and Pfizer achieved a breakthrough with their vaccine against COVID-19, the first-ever approved mRNA vaccine that demonstrates exceptionally high efficacy and good safety [85]. The mRNA encodes for the SARS-CoV-2 spike protein and is formulated in lipid nanoparticles [86,87]. Soon afterwards, a second SARS-CoV-2 lipid nanoparticle-encapsulated mRNA-based vaccine (mRNA-1273 from Moderna) was approved [88].

Similar to RNA vaccines, DNA vaccines provide an intrinsic adjuvant effect apart from encoding the antigen of interest. However, it has been shown that DNA vaccines trigger the AIM2 inflammasome rather than TLR9 [89,90].

The findings about the applications of the TLR agonists, their clinical status, and safety concerns are summarized in the Table 1 below.

## 8. Concluding Remarks

Taking into account the strengths and limitations of the different TLR ligands, immune responses triggered by TLR4 and TLR9 appear to be most beneficial for clinical application. TLR4 agonist containing adjuvants have been approved as part of several vaccines: Cervarix against cervical cancer, Shingrix against shingles and Fendrix against Hepatitis B. Similarly, a TLR9 agonist is included in another Hepatitis B vaccine, Heplisav-B. These adjuvants demonstrate a capacity for dose sparing and induce strong immune responses in immunocompromised individuals. TLR4 agonists have been tested as adjuvants in vaccines against infectious diseases, anti-allergic treatments and as immunotherapy against cancer. TLR9 agonists have also been investigated as vaccine components against infectious diseases and in anticancer immunomodulation. It is worth noting that TLR4 and TLR9 agonists were discovered first, which might explain their advanced development as vaccine adjuvants.

Although they have demonstrated some benefit as immunotherapy agents against cancer, AIDS and CFS/ME, TLR3 agonists also cause adverse effects in some individuals and have not yet been authorised for use in vaccines. TLR5 agonists have been tested in anti-tumour immunotherapy. Here, the novel vaccine against cancer, Mobilan, appears to be especially promising due to its creative design, which allows it to be effective regardless of TLR5 expression on the targeted cancer cells. TLR7 agonists, being small molecules, quickly disperse from the site of injection and induce a limited immune response, restricting their clinical application. However, introducing a depot of the adjuvant at the site of injection could circumvent this problem.

Other challenges remain within the field of TLR agonists for cancer immunotherapy. Tumour cells can evolve mechanisms to circumvent or counteract the immune defence through TLR signalling [100]. Activation of particular TLRs on cancer cells strengthens immune suppression by reducing cytotoxicity of immune cells, boosting the production of pro-inflammatory cytokines and triggering an aberrant form of tissue repair benefitting the tumour, which ultimately leads to tumour evasion from immune surveillance. A possible solution might be to target several PRR families, thereby enhancing protective responses in this context [101]. Synergy and crosstalk between PRR families, such as the synergy of TLR4 with NOD1 and NOD2 signalling to induce DC maturation, have already been demonstrated [101].

The current SARS-CoV-2 pandemic strikingly shows the ever-present need for fast and effective vaccine development. Remarkably, an enormous global effort has led to the licensure of several vaccines within one year. The TLR-activating RNA vaccines against COVID-19 have demonstrated the great potential of these PRRs as adjuvant targets. The ease of modifying the antigen insert along with their mass application against COVID-19 will likely accelerate licensure of future RNA-based vaccines against other pathogens.

In summary, TLR agonists have emerged as highly potent activators of innate immunity in a number of vaccine adjuvants and immunomodulatory agents against both infectious diseases and cancer. Their inclusion should be actively considered in novel vaccine design and the development of immunotherapeutic treatments. Furthermore, the choice of the delivery vehicle for the TLR ligands, combined with a careful formulation, are key in inducing the desired immune response while preventing adverse effects.

## Figures and Tables

**Figure 1 pharmaceutics-13-00142-f001:**
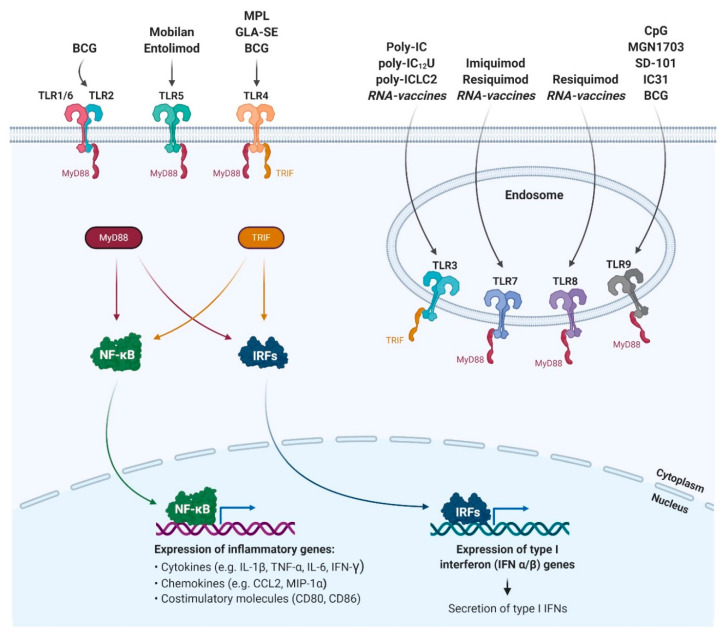
TLR-activating adjuvants and the induced signalling pathway. Cell surface TLRs (TLR1, TLR2, TLR4, TLR5, TLR6) and intracellular (endosomal) TLRs (TLR3, TLR7, TLR8, TLR9) are the targets of the illustrated adjuvants. Upon binding of the appropriate adjuvant by the LRR domain, TLRs recruit TIR domain-containing adaptor proteins like MyD88 and TRIF, which activate NF-κB signalling, MAP kinases (not shown) and IRFs) (e.g., IRF3, IRF7 mediated type I IFN responses). This initiates the expression of pro-inflammatory cytokines such as TNF-α, IL-6, IFN-γ or the type I interferons IFN-α and IFN-β. The TIR domain-containing adapter proteins in the figure are colour-coded to show whether the signalling of each TLR is MyD88- or TRIF-dependent. Abbreviations: TLR (Toll-like receptor), LRR (leucine-rich repeat), TIR (Toll-Interleukin receptor), MyD88 (myeloid differentiation factor 88), TRIF (TIR domain-containing adaptor protein inducing IFNβ), NF-κB (nuclear factor kappa-light-chain-enhancer of activated B cells), MAP (mitogen-activated protein), IRF (Interferon Regulatory Factor), IFN (Interferon), TNF-α (tumour necrosis factor-alpha), IL (Interleukin); Created with BioRender.com.

**Table 1 pharmaceutics-13-00142-t001:** Application, clinical trial or approval status, and safety profile of clinically used TLR agonists.

TLR	TLR Agonist	Application	Status	Safety Profile
TLR3	Poly-IC	Leukaemia and solid tumours	Phase I/II [20]	Severe toxic side effects in some patients (fever in 66%, transient elevation of serum glutamic-oxaloacetic transaminase and serum glutamic-pyruvic transaminase in 25%, minimal laboratory evidence of coagulation abnormalities in 59%, and hypersensitivity in 5%).
	Poly-IC12U	HIV type 1 infection	Phase I. [22]	No significant adverse events (AEs).
		CFS/ME	Phase III [24]	No significant AEs.
		Ovarian cancer and Peritoneal malignancies	Phase I ongoing (NCT01312389) [25]	N/A
	Poly-ICLC	Anaplastic brain cancers	Phase I/II [27]	No significant AEs.
		Ovarian cancer	Phase I [28]	Generally well tolerated.
TLR4	MPL	Herpes zoster	Approved vaccine Shingrix	Well tolerated, serious AEs are rare [91].
		Papillomavirus	Approved vaccine Cervarix	Localised site of injection reactogenicity reported due to the AS04 adjuvant [92].
		Hepatitis B	Approved vaccine Fendrix	Mild local and systemic AEs in up to 50% of vaccines [93].
		Allergic rhinitis	Approved immunotherapy Pollinex Quattro	Local reactogenicity in 5–6% of vaccinees and rare <0.7% systemic effects (rhinitis, breathing problems) [94].
	GLA-SE	Influenza	Phase II [37]	Generally well tolerated.
		Tuberculosis	Phase I [38]	Generally well tolerated.
		Soft tissue sarcoma	Phase I [39]	Generally well tolerated.
		Merkel cell carcinoma	First-in-human study [40]	Generally well tolerated.
TLR5	Mobilan	Prostate cancer	Phase I [43]	Satisfactory safety and tolerability.
	Entolimod	Advanced cancers	Phase I/investigational new drug, fast-track, and orphan drug status [45]	Some common toxicities observed but can be combined with other immunotherapeutic agents against cancer.
TLR7/8	Resiquimod	High-risk melanoma	Phase I [48]	Generally well tolerated.
TLR7	Imiquimod	Genital warts, superficial basal cell carcinoma, actinic keratosis	Approved cream Aldara	Local application site reactions, minimal systemic absorption [95].
		Malignant melanoma	Phase I [51]	Excellent safety profile for topical application.
		Breast cancer skin metastases	Phase II [52]	Transient mild to moderate local and systemic AEs.
TLR9	CpG 1018	Hepatitis B	Approved vaccines Heplisav-B (CpG 1018) and Engerix-B (CpG 7909)	Heplisav-B compared favourably to Engerix-B in both safety and efficacy [96]. Fatigue, headache in up to 25% of vaccines [97].
	CpG 7909	Malaria	Phase I [58]	Most related adverse events were mild or moderate, but 4 volunteers experienced severe systemic reactions and two were withdrawn from vaccinations due to adverse events.
	MGN1703	Metastatic colorectal carcinoma	Phase II [60]	AEs mild to moderate and limited to the injection-site.
	SD-101	Advanced melanoma	Phase Ib [64]	Injection-site reactions and transient, mild-to-moderate “flu-like” AEs.
	SD101	Low-grade non-Hodgkin lymphoma	Phase I/II [66]	No serious AEs.
	IC31	Tuberculosis	Phase I [98]	No serious adverse events were reported. Nine subjects (36%) presented with transient cardiovascular AEs.
TLR2, TLR4, TLR9	BCG	Bladder cancer	Phase II/III, comparison to Doxorubicin [99]	Treatment with BCG resulted in more frequent but not more severe adverse reactions than doxorubicin therapy.
TLR2, TLR4, TLR7, TLR9	BCG + 5% imiquimod	In-transit melanoma	Phase III [73]	Well tolerated.
TLR3, TLR7, TLR8	mRNA as vaccine platform with intrinsic adjuvanting effect	Prostate cancer	Phase I/IIa [83]	SAEs considered possibly treatment-related were reported in 2 (5%) of patients (urinary retention, hydronephrosis) Resolved after symptomatic and antibiotic treatment.
		SARS-CoV-2	Phase II/III [84,85]	Generally well tolerated.

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
