# Peer review of "TLR Agonists as Vaccine Adjuvants Targeting Cancer and Infectious Diseases"

_pharmaceutics, 2021, doi:10.3390/pharmaceutics13020142_

Round 1

Reviewer 1 Report

x

Luchner et al. provide a brief review on the current state of TLR agonists as vaccine adjuvants in the setting of cancer and infectious disease. Overall, the report is well-written and provides an evenly-balanced discussion of the various TLRs, their agonists and the existing clinical experience implementing them. The literature while not exhaustive, covers the bases and is up-to-date. The sole figure included provides sufficient information regarding TLR signaling pathways for novices to this area of research. I have no concerns for this work and would recommend acceptance in its current form.

Author Response

Thank you for the positive comments.

Reviewer 2 Report

This is an excellent review summarizing immune therapeutics to target the innate system for cancer immunotherapy. It is well written, comprehensive with good illustrations. Researchers in the field will definitely find this insightful. I feel this review can be further strengthened with a brief discussion on the challenges of some of these therapies in the clinic and a more comprehensive perspective of future directions.

Reviewer 3 Report

The Review “TLR agonists as vaccine adjuvants targeting cancer and infectious diseases” written by  Luchner et al. is an exceptional piece of work that emphasizes the TLR agonists as vaccine adjuvants. The authors have done a tremendous job: collecting data about all the different types of TLR related adjuvants and their role in various diseases. Although the article is well written and scientifically sound, There are few points if authors can add that will be interesting to read; those are as follows:

It would be good if the Authors can add some modern research regarding TLR with the historical prospects.

The authors should elaborate a little more on the last part of mRNA vaccines. If possible, write more details on how mRNA can act as a self adjuvant. If mRNA is self adjuvant, then what kind of approach is used in new Covid Vaccines.

Authors should also discuss DNA vaccines with or without these adjuvants.

Reviewer 4 Report

  • Authors should include more data on the CPG and more specifically to include all the known types/ classes (A, B, K,P), structural differences and distinguish the possible mechanisms of action between the classes
  • would clearly specify in the section of each adjuvant what is the cancer and the infectious diseases application
  • would suggest a comprehensive table with concise info of each adjuvant, possible mechanism of action, application and if known side effects/ safety concerns
  • abstract: the latter half is too detailed and not concise; also many abbreviations listed with full nomenclature
  • Historical perspective section: In the 3rd paragraph, too long mechanistic description is noted; should be moved.
